# Disseminating cells in human oral tumours possess an EMT cancer stem cell marker profile that is predictive of metastasis in image-based machine learning

Gehad Youssef[1], Luke Gammon[1], Leah Ambler[1], Sophia Lunetto[1], Alice Scemama[1], Hannah Cottom[2], Kim Piper[2], Ian C Mackenzie[1], Michael P Philpott[1], Adrian Biddle[1]*

[1]Blizard Institute, Barts and The London School of Medicine and Dentistry, Queen Mary University of London, London, United Kingdom; [2]Department of Cellular Pathology, Barts Health NHS Trust, London, United Kingdom

**Abstract** Cancer stem cells (CSCs) undergo epithelial-mesenchymal transition (EMT) to drive metastatic dissemination in experimental cancer models. However, tumour cells undergoing EMT have not been observed disseminating into the tissue surrounding human tumour specimens, leaving the relevance to human cancer uncertain. We have previously identified both EpCAM and CD24 as CSC markers that, alongside the mesenchymal marker Vimentin, identify EMT CSCs in human oral cancer cell lines. This afforded the opportunity to investigate whether the combination of these three markers can identify disseminating EMT CSCs in actual human tumours. Examining disseminating tumour cells in over 12,000 imaging fields from 74 human oral tumours, we see a significant enrichment of EpCAM, CD24 and Vimentin co-stained cells disseminating beyond the tumour body in metastatic specimens. Through training an artificial neural network, these predict metastasis with high accuracy (cross-validated accuracy of 87–89%). In this study, we have observed single disseminating EMT CSCs in human oral cancer specimens, and these are highly predictive of metastatic disease.

*For correspondence:
a.biddle@qmul.ac.uk

Competing interest: The authors declare that no competing interests exist.

## Editor's evaluation

This is a valuable study that convincingly demonstrates that quantification of EpCAM+/CD24+/Vimentin+ cells in the stroma of human oral cancers followed by machine learning algorithms can be used as a prognostic indicator of metastasis.

## Introduction

In multiple types of carcinoma, cancer stem cells (CSCs) undergo epithelial-mesenchymal transition (EMT) to enable metastatic dissemination from the primary tumour (*Biddle et al., 2011*; *Lawson et al., 2015*; *Liu et al., 2014*; *Ruscetti et al., 2016*). This model of metastatic dissemination has been built from studies using murine models and human cancer cell line models. However, this process has not been observed in human tumours in the in vivo setting, leading to uncertainty over the relevance of these findings to human tumour metastasis (*Bill and Christofori, 2015*; *Williams et al., 2019*). A key complication with efforts to study metastatic processes in human tumours is the inability to trace cell lineage. As cancer cells exiting the tumour downregulate epithelial markers whilst undergoing EMT, they become indistinguishable from the mesenchymal non-tumour cells surrounding the tumour

**eLife digest** When oral cancers metastasise – that is, when tumour cells invade other parts of the body – they typically do so by first colonizing the lymph nodes present in the neck. As this event significantly reduces chances of survival, oral cancer patients often have their neck lymph nodes removed to prevent the spread of the disease. However, this surgery carries risks and leads to longer hospital stays, stressing the need for better ways to predict which oral tumours will metastasise.

Evidence from lab-grown cells and mice studies suggest that, in oral cancer, metastasis occurs when some cells in the original tumour go through a process called the epithelial-mesenchymal transition (EMT for short). This transformation allows the cells to detach from the tumour and become invasive. However, it has so far been difficult to observe this process in actual human tumours; this is partly because cells undergoing EMT stop producing the proteins that scientists rely on to distinguish cancer and healthy cells.

To address this knowledge gap, Youssef et al. focused on three proteins: two tumour markers, EpCAM and CD24; and Vimentin, which is produced in greater quantities in the invasive mesenchymal state. Previous work had shown that a specific population of oral tumour cells can continue to express all three proteins even when adopting a mesenchymal identity through EMT.

Based on this knowledge, Youssef et al. hypothesised that tracking Vimentin, EpCAM and CD24 using fluorescence microscopy would allow them to identify metastasising cells in human samples. An analysis of over 12,000 images from 74 tumours obtained from surgeries revealed that, in the metastatic samples, the cells detaching from primary tumours were more likely to express these three proteins. Finally, Youssef et al. used these images to train a machine learning algorithm. When applied to data from new oral cancer patients, the programme was able to predict whether their tumours were likely to spread with 89% accuracy. If confirmed by further work, and in particular on larger samples, these findings could in the future help clinicians decide which patients with oral cancer would benefit the most from surgery to remove neck lymph nodes.

(*Li and Kang, 2016*). Therefore, once these cells detach from the tumour body and move away they are lost to analysis. Attempts have been made to use the retention of epithelial markers alongside acquisition of mesenchymal markers to identify cells undergoing EMT in human tumours (*Bronsert et al., 2014*; *Jensen et al., 2015*; *Puram et al., 2017*). However, these studies were limited to characterising cells undergoing the earliest stages of EMT whilst still attached to the cohesive body of the primary tumour.

EMT must be followed by the reverse process of mesenchymal-to-epithelial transition (MET) to enable new tumour growth at secondary sites, and therefore retained plasticity manifested as ability to revert to an epithelial phenotype is an important feature of metastatic CSCs (*Ocaña et al., 2012*; *Tsai et al., 2012*). We have previously demonstrated that a CD44$^{high}$EpCAM$^{low/-}$ EMT population can be separated from the main CD44$^{low}$EpCAM$^{high}$ epithelial population in flow cytometric analysis of oral squamous cell carcinoma (OSCC) cell lines and fresh tumour specimens (*Biddle et al., 2016*; *Biddle et al., 2011*). We identified retained cell surface expression of EpCAM (*Biddle et al., 2011*) and CD24 (*Biddle et al., 2016*) in a minority of cells that have undergone a full morphological EMT. Both EpCAM and CD24 were individually associated with enhanced ability to undergo MET, and thus are markers of EMT CSCs exhibiting retained plasticity. We therefore reasoned that retention of one or both of these markers may identify an important population of tumour cells that have undergone EMT and disseminated from the primary tumour in human tumour specimens, and are responsible for subsequent metastatic seeding. Here, we characterise the combined role of EpCAM and CD24 in marking a population of disseminating tumour cells in human OSCC specimens. Staining for EpCAM and CD24 alongside the mesenchymal marker Vimentin in over 12,000 imaging fields from 74 human tumours, stratified on metastatic status, identifies cells that have undergone EMT and disseminated into the stromal region surrounding metastatic primary tumours. Using an image-based machine learning approach, we show that the presence of these EMT CSCs in the tumour stroma is predictive of metastasis.

## Results

### Identification of human tumour cells that have undergone an EMT and disseminated into the surrounding stromal region

The retention of EpCAM expression in a sub-population of tumour cells that have undergone EMT raised the prospect that we may be able to identify these cells outside of the tumour body in human tumour specimens, as EpCAM is a specific epithelial marker that would not normally be found in the surrounding stromal region. In combination with EpCAM, we stained tumour specimens for CD24 as a second marker of plastic EMT CSCs, and Vimentin as a mesenchymal marker to identify cells that have undergone EMT. Notably, CD44 cannot be used as an EMT marker in the context of intact tissue as it requires trypsin degradation in order to yield differential expression in EMT and epithelial populations (*Biddle et al., 2013*; *Mack and Gires, 2008*). Vimentin, on the other hand, accurately distinguishes EMT from epithelial tumour cells in immunofluorescent staining protocols (*Biddle et al., 2016*). By combining EpCAM as a tumour lineage and EMT CSC marker, Vimentin as a mesenchymal marker, and CD24 as a plastic EMT CSC marker, we aimed to identify tumour cells that have undergone EMT and disseminated into the surrounding stromal region. For this, we developed a protocol for automated four-colour (three markers +nuclear stain) immunofluorescent imaging and analysis of entire histopathological slide specimens, to test for co-localisation of the three markers in each individual cell across each specimen.

To determine whether this marker combination identifies EMT CSCs, we initially tested the protocol on the CA1 OSCC cell line and an EMT CSC sub-line that is a derivative of this cell line (EMT-stem sub-line; *Biddle et al., 2016*). EpCAM$^+$Vim$^+$CD24$^+$ cells were greatly enriched in the EMT-stem sub-line, comprising 41% of the population, compared to 2.1% in the CA1 line (*Figure 1A, B and E*). Cells with this staining profile were absent from normal keratinocyte culture and cancer associated fibroblast culture (*Figure 1—figure supplement 1*). To test the specific role of EpCAM retention, we replaced EpCAM with a pan-keratin antibody against epithelial keratins. There was very little Pan-keratin$^+$Vim$^+$CD24$^+$ staining, and no enrichment for Pan-keratin$^+$Vim$^+$CD24$^+$ cells in the EMT-stem sub-line (*Figure 1C, D and E*). Therefore, whilst epithelial keratins are lost, EpCAM is retained in cells undergoing EMT and an EpCAM$^+$Vim$^+$CD24$^+$ staining profile can be used as a marker for EMT CSCs in immunofluorescent staining protocols.

Imaging the tumour body and adjacent stroma in sections of human OSCC specimens, we detected single cells co-expressing EpCAM, Vimentin and CD24 in the stromal region surrounding the tumour (*Figure 1F*), confirming that these cells can be detected in human tumour specimens. We next stratified 24 human primary OSCC specimens into 12 tumours that had evidence of lymph node metastasis or perineural spread, and 12 that remained metastasis free (*Supplementary file 1*), and stained them for EpCAM, Vimentin and CD24. Single cells co-expressing EpCAM, Vimentin and CD24 were abundant in the stroma surrounding metastatic tumours. This was not the case in non-metastatic tumours or normal epithelial regions (*Figure 2*, A-C). In contrast to EpCAM, pan-keratin staining did not identify cells in the stroma surrounding metastatic tumours (*Figure 2D*).

We developed an image segmentation protocol that separated the tumour body from the adjacent stroma, thus enabling each nucleated cell to be assigned to either the tumour or stromal region in automated image analysis (*Figure 2E*). Expression of EpCAM, Vimentin and CD24 was then analysed for every nucleated cell in every imaging field that included both tumour and stroma (3500 manually curated imaging fields across the 24 tumours). This enabled the proportion of each cell type in each region to be quantified (*Figure 2F*). EpCAM$^+$Vim$^+$CD24$^+$ cells were enriched in the stroma compared to the tumour body, and there was a much greater accumulation of EpCAM$^+$Vim$^+$CD24$^+$ cells in the stroma of metastatic tumours compared to non-metastatic tumours. Interestingly, this was not the case for EpCAM$^+$Vim$^+$CD24$^-$ cells, which were also enriched in the stroma but showed no difference between metastatic and non-metastatic tumours. Pan-keratin$^+$Vim$^+$CD24$^+$ cells were not detected.

To extend this analysis, we stained and imaged a further 59 tumour slides from 54 regions across 50 additional tumours, stratified on the same criteria. These displayed the same evidence of individual disseminating cells co-expressing EpCAM, Vimentin and CD24 in metastatic tumours only (*Figure 2G* and *Figure 2—figure supplements 1 and 2*). For these tumours, using a variation on the previous image segmentation protocol (*Figure 2—figure supplement 3A–D*), the proportion of EpCAM$^+$Vim$^+$CD24$^+$ and EpCAM$^+$Vim$^+$CD24$^-$ cells was quantified for each cell in over 9000 imaging fields at

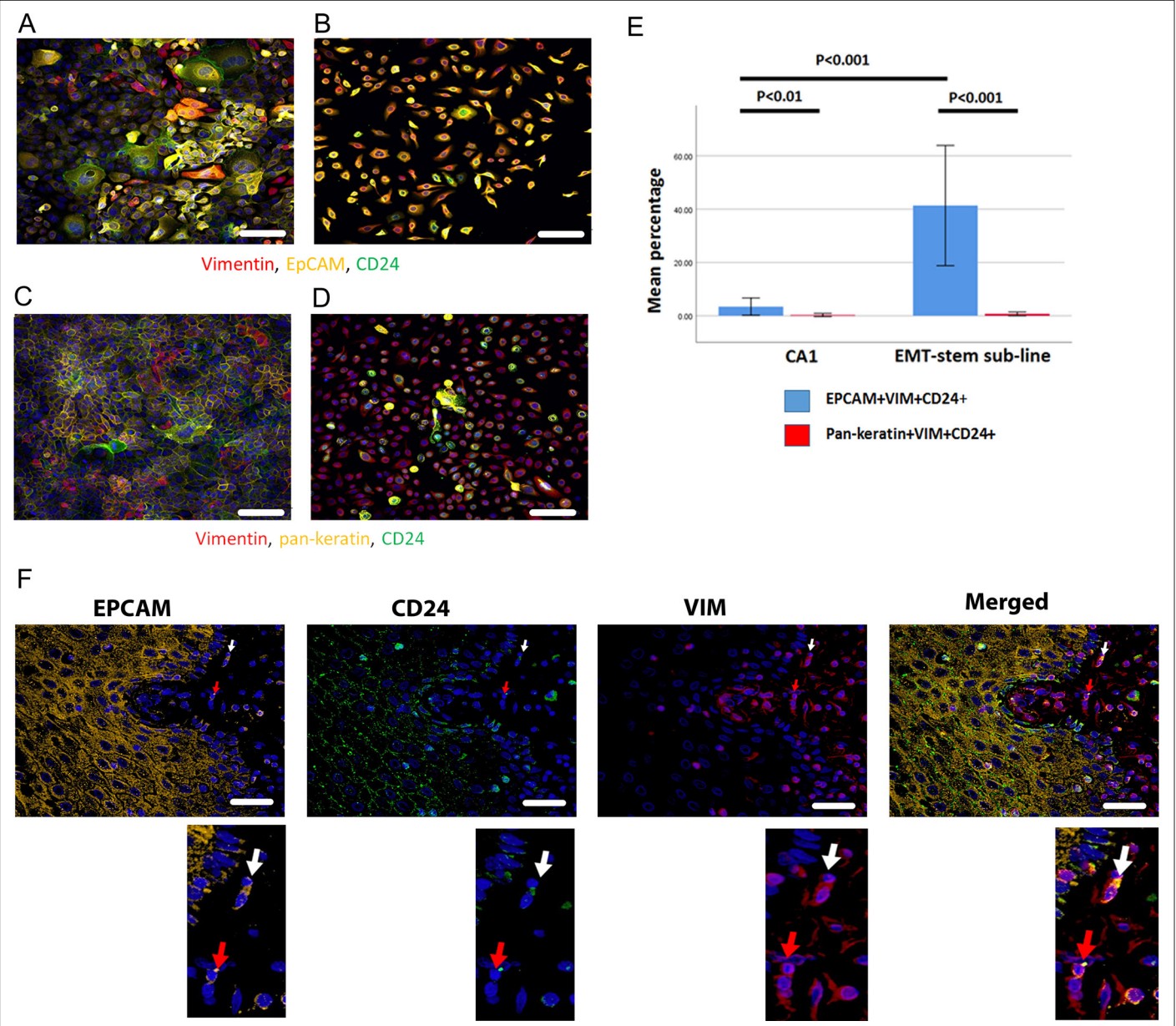

**Figure 1.** Immunofluorescent co-staining for EpCAM, Vimentin and CD24 identifies the EMT stem cell state. (**A–D**) Immunofluorescent staining for EpCAM, Vimentin and CD24 (**A, B**) and pan-keratin, Vimentin and CD24 (**C, D**) in the CA1 cell line (**A, C**) and the EMT-stem CA1 sub-line (**B, D**). (**E**) Quantification of the percentage of EpCAM⁺Vim⁺CD24⁺ and pan-keratin⁺Vim⁺CD24⁺ cells in the CA1 cell line and EMT-stem sub-line. Significance is obtained from a two-tailed student t-test. The graph shows mean +/-95% confidence interval. n=3. (**F**) Detection of EpCAM⁺Vim⁺CD24⁺ cells in the stroma surrounding an oral cancer tumour specimen. The white arrow highlights an EpCAM⁺Vim⁺CD24⁺ cell in the stroma. The red arrow highlights an EpCAM⁺Vim⁺CD24⁻ cell in the stroma. DAPI nuclear stain is blue. Below inset; enlargement of the highlighted cells for each marker. Scale bars = 100 μm.

The online version of this article includes the following figure supplement(s) for figure 1:

**Figure supplement 1.** EpCAM, Vimentin and CD24 immunofluorescent staining in the CA1 OSCC cell line (left), normal keratinocytes (centre) and oral cancer associate fibroblasts (right).

the tumour-stroma boundary (*Figure 2—figure supplement 3E*). Consistent with the previous set of tumours, only EpCAM⁺Vim⁺CD24⁺ cells were specifically enriched in the stroma of metastatic tumours.

To explore whether these EpCAM⁺Vim⁺CD24⁺ cells in the stroma may in fact be non-tumour cell types, we analysed a published scRNAseq dataset for human head and neck cancer (*Puram et al., 2017*). In this dataset, tumour and non-tumour cells were separated using bioinformatic techniques (principally inferred CNV and a 'tumour-epithelial' expression signature). Analysing this dataset for EpCAM,

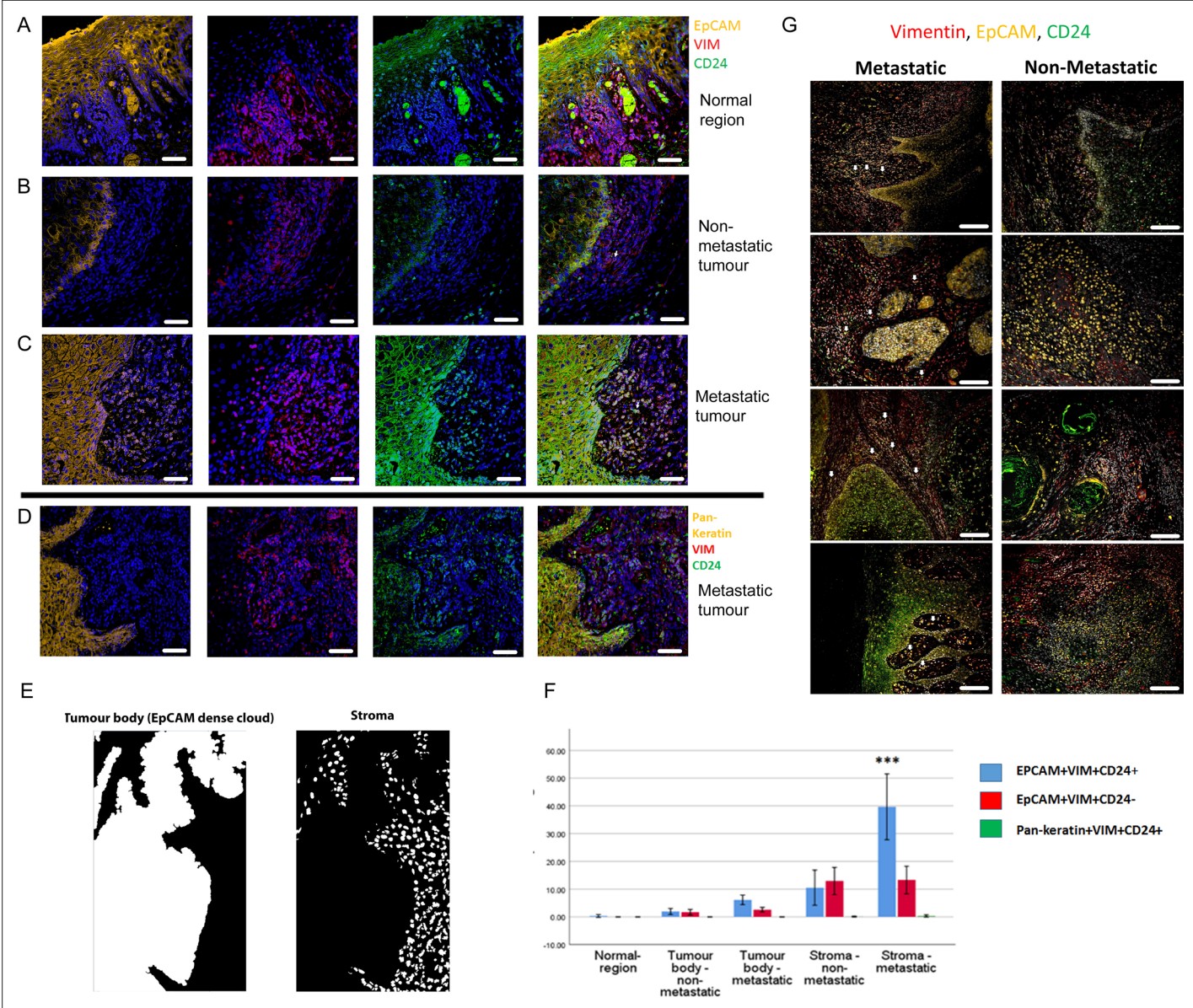

**Figure 2.** Enrichment of EpCAM⁺Vim⁺CD24⁺ cells in the stroma surrounding metastatic tumours. (**A–C**) Immunofluorescent four-colour staining of oral tumour specimens for EpCAM (yellow), Vimentin (red) and CD24 (green) with DAPI nuclear stain (blue). Representative imaging fields from a normal epithelial region (**A**), a non-metastatic tumour (**B**) and a metastatic tumour (**C**). (**D**) Staining of a metastatic tumour for pan-keratin, Vimentin and CD24. (**E**) Image segmentation was performed, with generation of an 'EpCAM dense cloud' to distinguish the tumour body from the stroma. Grey level intensities for EpCAM, Vimentin and CD24 were obtained for every nucleated cell in each imaging field. (**F**) Quantification of the percentage of EpCAM⁺Vim⁺CD24⁺, EpCAM⁺Vim⁺CD24⁻ and pan-keratin⁺Vim⁺CD24⁺ cells in normal region (epithelium distant from the tumour), tumour body, and stromal region from metastatic and non-metastatic tumours in the first cohort of specimens. A student t-test was performed comparing the mean percentage of EpCAM⁺Vim⁺CD24⁺ co-expressing cells in the metastatic stroma compared to the other fractions. *** signifies $p<0.001$. The graph shows mean +/-95% confidence interval. (**G**) Immunofluorescent four-colour staining of oral tumours from the second cohort of specimens, showing tumours with a range of invasive front presentations. White arrows highlight single EpCAM⁺Vim⁺CD24⁺ cells in the stroma. Scale bars = 100 µm.

The online version of this article includes the following figure supplement(s) for figure 2:

**Figure supplement 1.** The metastatic tumour fields from *Figure 2G*, shown with separate channels at the top and the merge below.

**Figure supplement 2.** The non-metastatic tumour fields from *Figure 2G*, shown with separate channels at the top and the merge below.

**Figure supplement 3.** Enrichment of EpCAM⁺Vim⁺CD24⁺ cells in the stroma surrounding metastatic tumours in the second cohort of specimens.

**Figure supplement 4.** Analysis of EpCAM, CD24, and Vimentin expression in a published head and neck cancer scRNAseq dataset (*Puram et al., 2017*).

Vimentin and CD24 co-expression, we found that 12% of tumour cells (267/2215) were EpCAM$^+$Vim$^+$CD24$^+$. In the non-tumour cells, only 0.8% (29/3687) were EpCAM$^+$Vim$^+$CD24$^+$ (*Figure 2—figure supplement 4*). Therefore, the observed EpCAM$^+$Vim$^+$CD24$^+$ cells in our tumour specimens are highly likely to be a tumour cell population. Indeed, use of EpCAM as a tumour lineage marker is specifically intended to exclude staining for stromal constituents. EpCAM is a specific epithelial marker, that is not expressed in stromal or immune cells – it is expressed exclusively in epithelia and epithelial-derived tumours (*Keller et al., 2019*).

These findings demonstrate that an EpCAM$^+$Vim$^+$CD24$^+$ staining profile marks tumour cells disseminating into the surrounding stroma, and that these cells are enriched specifically in metastatic tumours. The presence of disseminating tumour cells that express EpCAM but not CD24 did not correlate with metastasis. This highlights a requirement for the plasticity marker CD24, when identifying disseminating metastatic CSCs.

## Identification of EpCAM$^+$CD24$^+$Vim$^+$ CSCs enables clinical prediction using a machine learning approach

OSCC are an important health burden and one of the top ten cancers worldwide, with over 300,000 cases annually and a 50% 5-year survival rate. There is frequent metastatic spread to the lymph nodes of the neck; this is the single most important predictor of outcome and an important factor in treatment decisions (*Sano and Myers, 2007*). If spread to the lymph nodes is suspected, OSCC resection is accompanied by neck dissection to remove the draining lymph nodes, a procedure with significant morbidity. At presentation it is currently very difficult to determine which tumours are metastatic and this results in sub-optimal tailoring of treatment decisions. Accurate prediction of metastasis would therefore have great potential to improve clinical management of the disease to reduce both mortality and treatment-related morbidity. We sought to determine whether the EpCAM$^+$CD24$^+$Vim$^+$ staining pattern could be predictive of metastasis.

Starting with the EpCAM, Vimentin and CD24 immunofluorescence grey levels for each nucleated cell, we used a supervised machine learning approach to predict whether an imaging field comes from a metastatic or non-metastatic tumour (*Figure 3A*). As a benchmark we used the pan-keratin, Vimentin and CD24 immunofluorescence grey levels, as we hypothesised that pan-keratin would provide an inferior predictive value than EpCAM given that there was no dissemination of pan-keratin expressing cells in the stroma. A total of 3500 imaging fields containing 2,640,000 total nucleated cells from 24 tumour specimens were used in the machine learning task. We compared the performance accuracy (10-fold cross-validated F-score) of different machine learning classification algorithms. The best performing classifiers for EpCAM, Vimentin and CD24 were the artificial neural network (ANN) and support vector machine (SVM), with F1 accuracy scores of 91% and 87% respectfully (*Figure 3B*). For the ANN, the area under the curve (AUC) accuracy score was 87%, with 94% sensitivity and 82% specificity. Training with Pan-keratin, Vimentin and CD24 gave much worse prediction across all classifiers (*Figure 3C*). These findings demonstrate that, utilising a machine learning algorithm, staining for EpCAM, Vimentin and CD24 can predict metastatic status with high accuracy and may therefore have clinical utility.

To further investigate its utility for metastasis prediction, we applied our trained ANN to an independent cohort of tumours in a blinded analysis. We stained and imaged 59 tumour slides from 54 regions across 50 tumours, stratified on the same criteria as the previous cohort, for EpCAM, Vimentin and CD24. We conducted a blinded test of 10 fields of view from each slide, to determine whether the ANN trained using the previous 24 specimens could predict the metastatic status of these new specimens. Taking the majority prediction from the 10 fields of view from each slide, the ANN correctly predicted metastatic status for 54/59 slides (*Supplementary file 2*). Next, we determined whether the increased input available from this new cohort could yield improved predictive accuracy. Over 9000 imaging fields at the tumour-stroma boundary, containing over 8.5 million nucleated cells, were fed into a new ANN machine learning task. For this task, we recorded the predictive accuracy from the training and validation sets after each training round ('epoch'), which showed good alignment and an 89% accuracy score after 12 training epochs (*Figure 3D*). Interestingly, this is a similar accuracy to the previous ANN trained using 3500 imaging fields from 24 specimens, suggesting that feeding more imaging data into the ANN may only improve prediction up to a point.

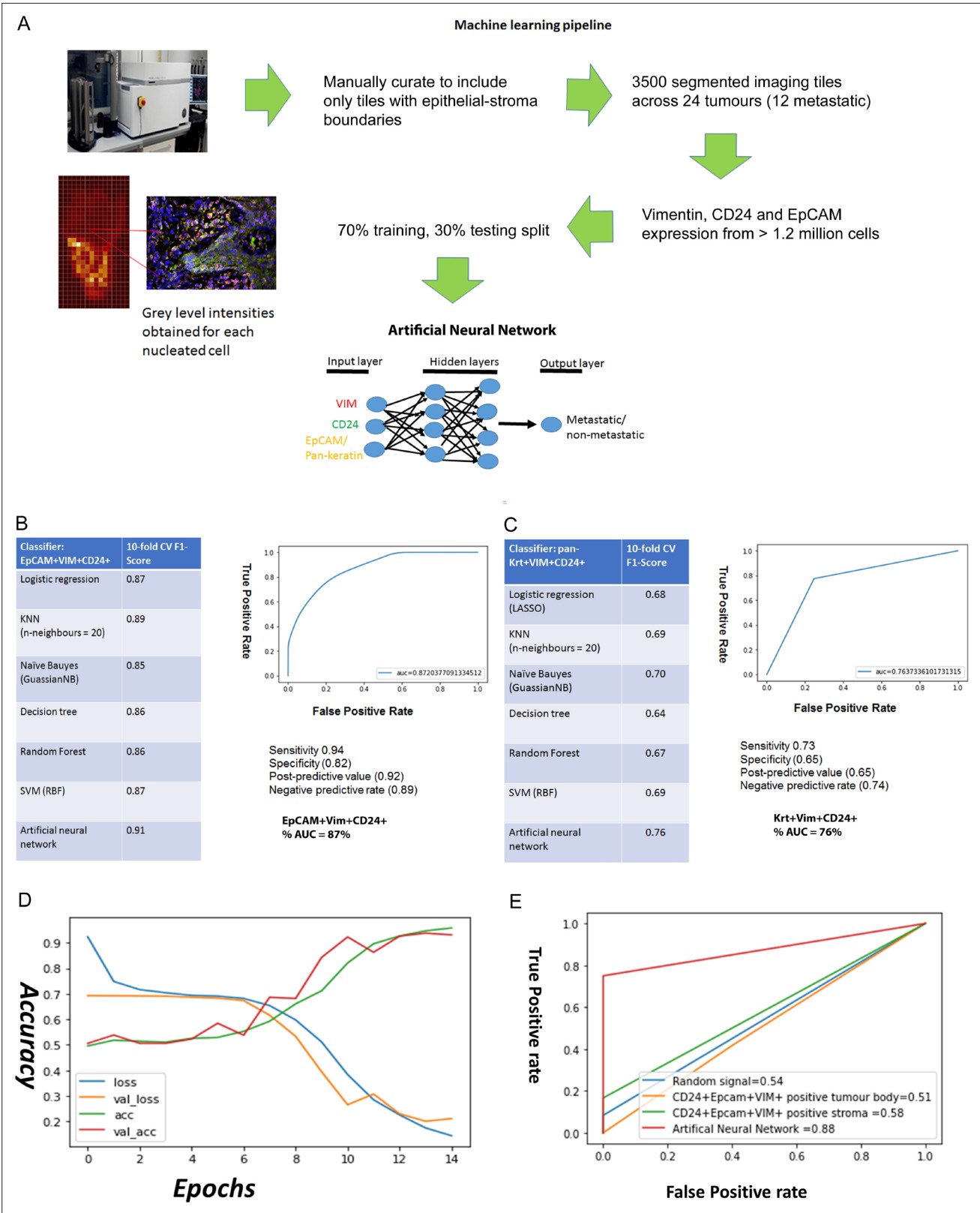

**Figure 3.** Predicting metastasis using EpCAM, Vimentin and CD24 immunofluorescent staining and a supervised machine learning approach. (**A**) Pipeline for machine learning based on grey level intensities for the three markers in tumour cohort 1. The training tiles were classified as coming from a metastatic or non-metastatic tumour. (**B, C**) Performance of EpCAM, Vimentin and CD24 (**B**) and pan-keratin, Vimentin and CD24 (**C**) in the supervised learning task on tumour cohort 1. The tables show the 10-fold cross-validation F1 scores of different machine learning classification

*Figure 3 continued on next page*

*Figure 3 continued*

algorithms. To the right of each table is a receiver-of-operator curve (ROC) showing the area under the curve (AUC) of the artificial neural network (ANN) classifier. (**D**) Performance of EpCAM, Vimentin and CD24 in the supervised learning task on tumour cohort 2. An ANN classifier was trained and tested on cohort 2, independently of tumour cohort 1. Accuracy and loss scores are displayed for the training set (green and blue lines) and the validation set (red and yellow lines) drawn from within this cohort, for 14 training epochs on the ANN classifier. (**E**) ROCs comparing accuracy of the image-trained ANN (red line) with ANNs trained using the number of EpCAM⁺Vim⁺CD24⁺ cells for each field of view from the tumour stroma (green line) and tumour body (yellow line). Training with random gaussian signals provided a baseline (blue line).

Finally, to test whether an imaging-based machine learning approach provides increased predictive accuracy over simpler approaches, we compared our ANN trained using image pixel intensity values to ANNs trained using summary data. These were trained using the CD24⁺EpCAM⁺VIM⁺-positive cell counts for each field of view, separated into tumour body and stroma using our imaging segmentation pipeline (*Figure 2E and F*). We used a dummy ANN fed with random gaussian signals as a baseline. The resulting ROC curves (*Figure 3E*) demonstrate that only the ANN trained using image pixel intensity values has predictive utility in our cohort.

To our knowledge, this is the first time immunofluorescent staining of human tumour tissue specimens has been used in a machine learning pipeline for clinical prediction. Previous studies using cytokeratin immunohistochemistry, clinicopathological data and serum biomarkers for clinical prediction via machine learning have achieved AUCs of 75% in breast cancer (*Tseng et al., 2019*), 80% in OSCC (*Bur et al., 2019*), and 82% in colorectal cancer (*Takamatsu et al., 2019*).

## Discussion

The role of EMT in tumour dissemination has long been debated but, lacking evidence of cells undergoing EMT whilst disseminating from human tumours in vivo, this role has had to be inferred from mouse models and human cell line models. Here, through applying our understanding of EMT cancer cell heterogeneity and markers for EMT CSCs, we have identified EMT CSCs disseminating from the primary tumour in human pathological specimens. Importantly, the presence of these disseminating stem cells is strongly correlated with tumour metastasis. Using an image-based machine learning approach, we have demonstrated the ability to predict metastasis with high accuracy through staining for EMT CSCs.

A partial EMT state has previously been identified in an OSCC scRNAseq dataset; this state retained epithelial gene expression alongside expression of mesenchymal genes, and was correlated with nodal metastasis and adverse pathological features (*Puram et al., 2017*). Now, using immunofluorescent staining for EMT CSCs that retain the epithelial marker EpCAM alongside the mesenchymal marker Vimentin and the CSC plasticity marker CD24, we have identified single EMT CSCs disseminating into the stroma surrounding oral tumours. However, epithelial keratins are not retained. We have also shown that retention of EpCAM is not on its own sufficient alongside Vimentin to mark disseminating EMT CSCs that correlate with metastasis. There is a requirement for CD24, which we have previously shown to be a plasticity marker within the EMT population even when driven into full morphological EMT under TGFβ treatment (*Biddle et al., 2016*). This suggests that the EMT CSC state may be more complex than a simple coalescence of epithelial and mesenchymal characteristics.

Image-based machine learning approaches employ a greater range of inputs to develop their predictions, and therefore often achieve greater predictive accuracy than machine learning approaches that use simpler data inputs. This comes at the cost of increased computing requirements and reduced ability to decode the learning process (the so-called 'black box'), but in our study the image-based approach proved essential to achieve high predictive accuracy. The lack of improvement in predictive accuracy when trained on a collection of 9000 images compared to 3500 suggests a limitation of the ANN, and a possible role for more sophisticated neural network architectures such as convolutional neural networks (CNNs). Whereas the ANN is trained using intensity values for each pixel, a CNN uses the raw image for training. CNNs require many more images for training, but can reach higher levels of accuracy. To support the conclusions of this study, a further blinded cohort and a greater number of tumour specimens would be valuable. This may also enable the application of a CNN to the imaging data to see if it can achieve even higher predictive accuracy than that reported here.

We have shown that immunofluorescent antibody co-staining for EpCAM, Vimentin and CD24 can separate disseminating EMT CSCs from the stromal content of human tumours, a challenge which has confounded previous attempts to develop a predictive EMT signature (*Tan et al., 2014*). We also show that EpCAM$^+$Vim$^+$CD24$^-$ tumour cells in the stroma do not correlate with metastasis, and therefore the clinically predictive utility of tumour cell staining in the stroma can be isolated specifically to the EpCAM$^+$Vim$^+$CD24$^+$ EMT CSCs. This highlights the value of using techniques that give single cell resolution, enabling isolation of the signal to the specific cell type of interest within a highly heterogeneous cellular environment. An important strength of our study has been the ability to look at the single cell level in an automated fashion across thousands of fields of view from human tumours, enabling us to observe and quantify human tumour cells disseminating into the surrounding tissue. This in turn enabled us to identify single disseminating EMT CSCs, and use these to train an ANN to predict metastasis.

# Methods

## Key resources table

| Reagent type (species) or resource | Designation | Source or reference | Identifiers | Additional information |
|---|---|---|---|---|
| Cell line (*Homo sapiens*) | CA1 oral cancer line | Generated in house through explant culture of a human tumour specimen | | |
| Cell line (*Homo sapiens*) | Cancer associated fibroblasts, primary culture | Generated in house through explant culture of a human tumour specimen | | |
| Cell line (*Homo sapiens*) | N/TERT non-transformed epidermal keratinocyte cell line. hTERT immortalised | *Smits et al., 2017* | | |
| Biological sample (*Homo sapiens*) | FFPE blocks of human oral cancer specimens | Barts Health NHS Trust diagnostic archive | | |
| Antibody | IgG2a mouse monoclonal CD24 antibody, clone ML5 | BD Bioscience | | 1:100 |
| Antibody | IgG rabbit recombinant monoclonal EpCAM antibody | Abcam | EPR20532-225 | 1:100 |
| Antibody | IgG1 mouse monoclonal Vimentin antibody, clone V9 | Dako | | 1:100 |
| Antibody | IgG rabbit polyclonal wide spectrum cytokeratin antibody | Abcam | Ab9377 | 1:100 |
| Antibody | Anti-mouse IgG2 Alexa Fluor 488 secondary antibody | Thermo Fisher | | 1:500 |
| Antibody | Anti-rabbit IgG Alexa Fluor 555 secondary antibody | Thermo Fisher | | 1:500 |
| Antibody | Anti-mouse IgG1 Alexa Fluor 647 secondary antibody | Thermo Fisher | | 1:500 |

*Continued on next page*

| Reagent type (species) or resource | Designation | Source or reference | Identifiers | Additional information |
|---|---|---|---|---|
| | | *Continued* | | |
| Chemical compound, drug | DAPI nuclear dye, kept as a 1 mg/ml stock in DMSO | Sigma | | 1:1000 |
| Software, algorithm | GE developer toolbox | GE | | |
| Software, algorithm | Skikit-learn Python 3.6 libraries | *Pedregosa et al., 2011* | | |
| Software, algorithm | Tensorflow/Keras framework | https://www.tensorflow.org/api_docs/python/tf/keras/models | | |

## Cell culture

The CA1 OSCC cell line and oral cancer associated fibroblasts were both previously derived in our laboratory, from separate biopsies of OSCC of the floor of the mouth. The EMT-stem sub-line was derived as a single cell clone from the CA1 cell line (*Biddle et al., 2016*). Normal keratinocytes were the N/TERT hTERT-immortalised epidermal keratinocyte cell line (*Smits et al., 2017*). Cell culture was performed as previously described (*Biddle et al., 2011*). Cell removal from adherent culture was performed using 1 x Trypsin-EDTA (Sigma, T3924) at 37 °C.

## Immunofluorescent staining of cell lines and tumour tissue sections

Tumour specimens were obtained from the pathology department at Barts Health NHS Trust, with full local ethical approval and patients' informed consent. Clinicopathological details for tumour cohort 1 are in *Supplementary file 1*, and clinicopathological details for tumour cohort 2 are in *Supplementary file 2*. Tumour cohort 1 consisted of 24 tumours. Tumour cohort 2 consisted of 54 distinct tumour blocks across 50 tumours. In addition, five tumour blocks were included twice (with two different sections). This made a total of 59 slides for cohort 2. From the tumour cohort 2 cases listed in *Supplementary file 2*, tumours 25, 28, 29, 32 were included as two separate blocks. Tumours 21, 29, 32, 33, 43 were included twice from the same block.

Sections of formalin fixed paraffin embedded (FFPE) archival specimens were dewaxed by clearing twice in xylene for 5 min then gradually hydrating the specimens in an alcohol gradient (100%, 90%, 70%) for 3 min each. The sections were then washed under running tap water before immersing the slides in Tris-EDTA pH9 for antigen retrieval using a standard microwave at high power for 2 min and then 8 min at low power.

Four-colour immunofluorescent staining was performed by firstly staining the membranous proteins prior to the permeabilisation and blocking steps. The sections were incubated with an IgG2a mouse monoclonal CD24 antibody (clone ML5, BD Bioscience) and IgG rabbit recombinant monoclonal EpCAM antibody (EPR20532-225, Abcam) in PBS overnight at 4 °C (1/100 dilution). The sections were then washed three times in PBS and incubated for 1 hr at room temperature with anti-mouse IgG2 Alexa Fluor 488 and anti-rabbit IgG Alexa Fluor 555 secondary antibodies (1/500 dilution). The sections were then washed in PBS and permeabilised with 0.5% triton-X in PBS for 10 min followed by blocking for 1 hr with blocking buffer (3% goat serum, 2% bovine serum albumin in PBS). The sections were then incubated with an IgG1 mouse monoclonal Vimentin antibody (clone V9, Dako) and (optionally, in place of EpCAM) IgG rabbit polyclonal wide spectrum cytokeratin antibody (ab9377, Abcam) overnight at 4 °C in blocking buffer (1/100 dilution). After washing with PBS, the sections were incubated with anti-mouse IgG1 Alexa Fluor 647 antibody and (optionally) anti-rabbit IgG Alexa Fluor 555 for 1 hr at 4 °C (1/500 dilution). After washing three times with PBS, cell nuclei were stained with DAPI (1/1000 dilution in PBS) for 10 min.

For cell line staining, cells were fixed in 4% PFA for 10 min then washed with PBS. Staining was performed in the same manner as described above, however permeabilisation was performed with 0.25% Triton-X for 10 min and DAPI incubation was reduced to 1 min.

## Quantifying the abundance of stained sub-populations in cell lines and tumour tissue sections

Imaging of the stained slides was performed using the In Cell Analyzer 2200 (GE), a high content automated fluorescence microscope with four-colour imaging capability. The slides were imaged at x20 and x40 magnification. An image segmentation protocol was developed to extract grey level intensities corresponding to EpCAM, Vimentin and CD24 expression for every DAPI stained nucleated cell in the tumour body and the adjacent stroma separately. Segmentation was performed using the Developer Toolbook software (GE). As shown in *Figure 2E* and *Figure 2—figure supplement 3A–D*, an 'EpCAM dense cloud' or 'Vimentin dense cloud' was generated to isolate individual nucleated cells in the tumour body from the adjacent stroma and analyse them separately.

Grey level intensities obtained from the imaging analysis were processed in the following way. Firstly, the median number of nucleated cells was calculated and imaging fields with fewer than 20% of the median nucleated cells were excluded from the analysis pipeline. The folded edges of a specimen were also excluded. The median grey level intensity of the FITC, CY3 and CY5 fluorescence channels corresponding to CD24, EpCAM and Vimentin expression were computed for the negative control stained slides. A nucleated cell was deemed to have positive CD24, EpCAM or Vimentin expression if its grey level intensity exceeded the background threshold value (1.5 x median grey level intensity of negative control slide) for the FITC, CY3 and CY5 channels, respectively. If a nucleated cell surpassed the background threshold for all three fluorescence channels it was termed a triple positive cell (CD24$^+$EpCAM$^+$Vim$^+$) and denoted with 1 and if this criteria was not met the nucleated cell was denoted with a 0. For EpCAM$^+$Vim$^+$CD24$^-$ cells (termed double positive), the nucleated cell must exceed the background threshold for the CY3 and CY5 channels but not the FITC.

The scRNAseq dataset (*Puram et al., 2017*) was analysed using a threshold (median or quartile) using the normalised count expression for EpCAM, CD24 and Vimentin for each cell.

## Machine learning for prognostic prediction using immunofluorescent staining data

A dataset was created of a pool of 2,640,000 nucleated cells across 3500 imaging fields from 24 tumour specimens (12 with lymph node metastasis or perineural spread, and 12 without) (cohort 1) or 8,563,000 nucleated cells across 9200 imaging fields from 59 tumour specimens (29 with lymph node metastasis or perineural spread, and 30 without) (cohort 2). The background threshold for the FITC, CY3 and CY5 channels was subtracted from the grey level intensities for each nucleated cell. The supervised machine learning task was to classify each imaging field into whether it belonged to a metastatic or non-metastatic tumour.

The dataset was stratified into a training and validation cohort in a 70%:30% ratio using a random seed split. Supervised machine learning approaches were implemented using the skikit-learn Python 3.6 libraries (*Pedregosa et al., 2011*) and Tensorflow/Keras framework (https://www.tensorflow.org/api_docs/python/tf/keras/models). Hyper-parameter optimisation was performed by an exhaustive grid search and computed on Apocrita, a high performance cluster (HPC) facility at Queen Mary University of London (http://doi.org/10.5281/zenodo.438045). To further minimise overfitting, 10-fold cross-validation was performed and the mean accuracy metric, F1 score, was obtained for each learning iteration. Receiver-of-operator (ROC) curves and the area-under the-curve (AUC) were computed for the optimum supervised learning algorithm. Supervised approaches used were logistic regression, support vector machines (*Smola and Schölkopf, 2004*), Naïve Bayes (*Zhang, 2005*), K-Nearest Neighbours (*Bentley, 1975*), decision trees (*Dumont et al., 2009*), and artificial neural networks (*Rumelhart et al., 1986*).

For the blinded analysis of tumour cohort 2 using the ANN trained on tumour cohort 1, the identity of the tumours was withheld by the pathologists at Barts Health NHS Trust and thus both the model and research team developing the model were blind to the clinical ground truth for the new imaging fields. For each new tumour, a pool of 10 imaging fields were used to make predictions using the previously trained ANN. A majority vote for the 10 imaging fields was obtained and each tumour was assigned either a metastatic or non-metastatic identity by our model. The clinical ground truth was then provided to us by the pathologists to confirm the accuracy of the predictions.

## Acknowledgements

We thank Ryan O'Shaughnessy, Sarah Marzi, and Jan Soetaert for technical assistance and discussion. Gehad Youssef and Adrian Biddle were supported by Animal Free Research UK, as part of the Animal Replacement Centre of Excellence at Queen Mary University of London. Leah Ambler was supported by Oracle Cancer Trust. Alice Scemama was supported by the National Centre for the 3Rs (NC3Rs). Adrian Biddle is a member of the Barts Centre for Squamous Cancer.

## Additional information

### Funding

| Funder | Grant reference number | Author |
|---|---|---|
| Animal Free Research UK | | Gehad Youssef<br>Michael P Philpott<br>Adrian Biddle |
| Oracle Cancer Trust | | Leah Ambler<br>Adrian Biddle |
| National Centre for the Replacement Refinement and Reduction of Animals in Research | NC/S001573/1 | Alice Scemama<br>Adrian Biddle |

The funders had no role in study design, data collection and interpretation, or the decision to submit the work for publication.

### Author contributions

Gehad Youssef, Conceptualization, Formal analysis, Investigation, Writing - original draft; Luke Gammon, Leah Ambler, Sophia Lunetto, Alice Scemama, Investigation, Methodology; Hannah Cottom, Resources, Data curation, Investigation; Kim Piper, Resources, Data curation; Ian C Mackenzie, Conceptualization, Supervision; Michael P Philpott, Supervision, Funding acquisition; Adrian Biddle, Conceptualization, Supervision, Funding acquisition, Investigation, Writing - original draft, Project administration, Writing - review and editing

### Author ORCIDs

Luke Gammon http://orcid.org/0000-0002-1233-2665
Michael P Philpott http://orcid.org/0000-0002-1255-4612
Adrian Biddle http://orcid.org/0000-0003-4371-9720

### Ethics

Human subjects: Archival human specimens and associated de-identified clinical data was accessed under UK HRA approval with REC ref 18/WM/0326.

### Decision letter and Author response

Decision letter https://doi.org/10.7554/eLife.90298.sa1
Author response https://doi.org/10.7554/eLife.90298.sa2

## Additional files

### Supplementary files

• Supplementary file 1. Tumour characteristics for the 12 metastatic tumours (tumours 1–12) and 12 non-metastatic tumours (tumours 13–24) that were sectioned and stained for EpCAM, CD24 and Vimentin in the first cohort.

• Supplementary file 2. Clinical details and blinded analysis outcomes for second tumour cohort.

• MDAR checklist

## Data availability

There are no sequencing datasets associated with this study. Publicly available packages used to analyse immunofluorescent images are listed in the methods section.

The following previously published dataset was used:

| Author(s) | Year | Dataset title | Dataset URL | Database and Identifier |
|---|---|---|---|---|
| Puram SV, Tirosh I, Parikh AS, Patel AP, Yizhak K, Gillespie S, Rodman C, Luo CL, Mroz EA, Emerick KS | 2017 | Single cell RNA-seq analysis of head and neck cancer | https://www.ncbi.nlm.nih.gov/geo/query/acc.cgi?acc=GSE103322 | NCBI Gene Expression Omnibus, GSE103322 |

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
