## [Editor Report]

This is a valuable study that convincingly demonstrates that quantification of EpCAM+/CD24+/Vimentin+ cells in the stroma of human oral cancers followed by machine learning algorithms can be used as a prognostic indicator of metastasis.

---

## [Decision Letter]

**Decision letter after peer review:**

[Editors’ note: the authors submitted for reconsideration following the decision after peer review. What follows is the decision letter after the first round of review.]

Thank you for submitting your work entitled "Disseminating cells in human tumours acquire an EMT stem cell state that is predictive of metastasis" for consideration by *eLife*. Your article has been reviewed by 3 peer reviewers, one of whom is a member of our Board of Reviewing Editors, and the evaluation has been overseen by a Senior Editor. The reviewers have opted to remain anonymous.

Our decision has been reached after consultation between the reviewers. Based on these discussions and the individual reviews below, we regret to inform you that your work will not be considered further for publication in *eLife*.

This manuscript was reviewed by experts in the areas of cancer stem like cells, EMT events and pathology. Overall, all of the reviewers were intrigued by the concepts underlying this paper. However, as indicated in the reviews, it was felt that the work is not yet ready for publication in *eLife*. Specifically, it seems that the work is validating the existence of an EMT-stem like population, whilst also attempting to formulate a clinical prognostic application for the existence of these cells. However, the function of these cells as metastatic drivers requires further exploration. Moreover, the pathological assessments must be improved upon. We hope that these comments are helpful.

*Reviewer #1 (General assessment and major comments (Required)):*

This manuscript follows previous studies describing the existence of a subpopulation of mesenchymal like cells (expressing Vimentin) that also express EpCAM and/or CD24 concomitant with the ability to of undergo MET. These subpopulations appear to exist within oral squamous cell carcinoma (OSCC) cell lines and within primary tissues. The paper demonstrates that CD24 expression is requisite for plasticity and suggests that the presence of CD24+/EpCAM+/VIM+ cells in the stroma of OSCC tumors may be indicative of metastasis. Some whole genome transcriptome analysis was also done to determine differences between bulk, EMT restricted and EMT stem populations. Overall, the notion that specific cells have the plasticity needed to move between epithelial and mesenchymal states is intriguing, and the presumption that these cells contribute to metastasis seems logical. However, the work is still rather preliminary. Accordingly, it is difficult to make solid conclusions regarding the prognostic utility of this state or of what may regulate it.

The study uses a very small sample size (24 patients) for the test and validation cohorts. The study should be expanded to use a different set of patient samples for test and validation sets. Moreover, the utility of the stem-EMT signature should be tested using multivariate analyses.

In figure 4, it looks like CD24 is positive in the bulk of tumors (regardless of stage) and in skin. Is this specific? Also, there appear to be VIMENTIN/EPCAM/CD24 positive cells in the bulk of non-metastatic tumours. Can this be seen using sequencing? Overall, the images as presented are not overly convincing.

EMT stem versus restricted signatures should be validated using additional models. Also, greater evidence is required to determine how these cell fractions may differ. Are they sitting in different epigenetic states? Can trajectories be detected in human cancers, using single cell sequencing, for example? Finally, do they have different metastatic potentials?

*Reviewer #2 (General assessment and major comments (Required)):*

The authors tackle the important and intractable question of the mismatch between the primacy of EMT in cell culture studies versus the rarity with which EMT is morphologically apparent in resected tumour tissues.

The early part of the study is convincing and well conducted, with identification of subpopulations of EMT cells with the ability to undergo MET, and associated marker profiles in flow cytometry.

They then develop an impressive multiplex assay for the identification of cells with the same profile in resected tumour material- a really promising approach bringing molecular findings into the context of primary tumour tissue.

The major issue that I have is in the application of this assay to tissues, and the subsequent AI analysis. Only one example of the putative invading population is shown (Figure 4C) and the stromal 'infiltrative' subpopulation is adjacent to a very flat and 'pushing' tumour/stroma boundary, with no apparent budding into the stroma. This would need to be addressed with several more examples and high-magnification H&E images. Furthermore, this is a major claim- namely that occult infiltrating EMT cells are commonly encountered in peritumoural stroma but can only be differentiated from somatic stroma by multiplex immunofluorescence- and it needs major evidence to back it up. What do these cells look like on H&E? Are they mesenchymal in their appearances on H&E? Can they be conclusively differentiated from other stromal constituents (eg myofibroblasts, plasma cells) immunohistochemically and/or morphologically? It could be that the power to predict metastatic status power is related to somatic stromal factors rather than EMT.

The AI prediction of metastatic status is compelling, but this fundamental point would need to be persuasively addressed in order to support the author's major claims.

I do not feel qualified to comment upon the AI strategies used later in the study.

*Reviewer #3 (General assessment and major comments (Required)):*

The article by Youssef G et al., focused on developing a Machine Learning system to use immunofluorescence data to detect metastatic cells in tumor stroma, which might be responsible for metastasis in case of OSCC. To detect single cell in the transition of EMT to MET they focused on EMT-Stem cells rather than only EMT phenotypes. They have shown that retention of epithelial marker EpCAM and stem cell marker CD24 and upregulation mesenchymal marker Vimentin can identify disseminating EMT stem cells in the tumor stroma. It is very well presented, well written and has high implication.

Comments to improve:

1. Strongly recommended to add the distribution of tumor status vs. proposed marker expression pattern. That is to show the distribution of EpCAM, CD24, Vimentin +/- in metastatic vs. other tumor status as mentioned in Supplementary figure 2. This might help you to establish these markers combination to follow a pattern in disease progression.

2. In all cell and tissue images add the scale.

3. For figure 3f, show enlarged picture of the single cell staining on the inset or add a separate panel to show only single cell staining.

4. Figure 4, the panel name or the font is too small to read, enlarge the font size (a, b, c, d, f).

5. Same problem with figure 6a, font size too small. In addition, in the heat maps, is it possible to add cluster names horizontally? Also for figure 6c, the cluster names are too small.

6. The EMT sub-populations are not associated with a spectrum of epithelial/mesenchymal genes expression (supplementary figure 5). The explanation is not very clear.

[Editors’ note: further revisions were suggested prior to acceptance, as described below.]

Thank you for submitting your article "Disseminating cells in human oral tumours acquire an EMT cancer stem cell state that is predictive of metastasis" for consideration by *eLife*. Your article has been reviewed by 3 peer reviewers, and the evaluation has been overseen by a Reviewing Editor and Richard White as the Senior Editor.

Essential revisions:

1) The authors should test the staining-AI protocol use another independent blinded cohort.

2) Please add clinical data for all patient samples.

3) The authors should edit manuscript to ensure clarity.

4) The need of the AI approach should be better articulated.

*Reviewer #2 (Recommendations for the authors):*

Here author used cell lines to standardize their protocol and used a test patient set to validate these markers' potential. Here they used a set of specimen slides with known histopathology and checked the marker's potential to predict the tumor status and how efficient the technology was.

– The major criticism is that the authors did not test the AI to determine the tissue status from a blind sample set. They need to show the efficiency of the AI to determine tumor, non-tumor, or metastasis status from a pool of specimens and match the AI predictions with the standardized histopathology data and their four markers IF protocols. What is the level of detection efficiency and what is the sensitivity and specificity of the technology?

– While validating a new marker for OSCC, even with a small sample set, it is strongly recommended to show metastasis potential in vivo.

*Reviewer #3 (Recommendations for the authors):*

The paper has numerous technical issues which will have to be addressed before accepting it. First and foremost, I am baffled why the fibroblasts in Figure S1 are EpCAM positive. If this is not an error, I can't understand the rest of the manuscript.

Beyond that, I find many of the figures challenging to the eye and in the case of Figure 2F, making key evidence hard to understand. Perhaps because of that, I couldn't see anywhere explaining properly how tumour "mask" was applied (Figure 2E?!).

Figure 3 also needs revision: none of the terms is described anywhere. For non-expert biologists reading *eLife* terms such as Epochs are entirely incomprehensible.

---

## [Author Response]

[Editors’ note: the authors submitted for reconsideration following the decision after peer review. What follows is the decision letter after the first round of review.]

Our decision has been reached after consultation between the reviewers. Based on these discussions and the individual reviews below, we regret to inform you that your work will not be considered further for publication in eLife.This manuscript was reviewed by experts in the areas of cancer stem like cells, EMT events and pathology. Overall, all of the reviewers were intrigued by the concepts underlying this paper. However, as indicated in the reviews, it was felt that the work is not yet ready for publication in eLife. Specifically, it seems that the work is validating the existence of an EMT-stem like population, whilst also attempting to formulate a clinical prognostic application for the existence of these cells. However, the function of these cells as metastatic drivers requires further exploration. Moreover, the pathological assessments must be improved upon. We hope that these comments are helpful.Reviewer #1 (General assessment and major comments (Required)):This manuscript follows previous studies describing the existence of a subpopulation of mesenchymal like cells (expressing Vimentin) that also express EpCAM and/or CD24 concomitant with the ability to of undergo MET. These subpopulations appear to exist within oral squamous cell carcinoma (OSCC) cell lines and within primary tissues. The paper demonstrates that CD24 expression is requisite for plasticity and suggests that the presence of CD24+/EpCAM+/VIM+ cells in the stroma of OSCC tumors may be indicative of metastasis. Some whole genome transcriptome analysis was also done to determine differences between bulk, EMT restricted and EMT stem populations. Overall, the notion that specific cells have the plasticity needed to move between epithelial and mesenchymal states is intriguing, and the presumption that these cells contribute to metastasis seems logical. However, the work is still rather preliminary. Accordingly, it is difficult to make solid conclusions regarding the prognostic utility of this state or of what may regulate it.

We are grateful that this reviewer has appreciated the importance of the research question and the value of our approach to it. The reviewer indicated that our manuscript needed further development, and detailed the areas requiring development in their comments below. We have now substantially revised our manuscript to address these comments, and will address each in turn.

The study uses a very small sample size (24 patients) for the test and validation cohorts. The study should be expanded to use a different set of patient samples for test and validation sets. Moreover, the utility of the stem-EMT signature should be tested using multivariate analyses.

We initially performed this experiment using 10 samples, and saw a large difference in staining between the metastatic and non-metastatic specimens. Based on this large difference, we determined that a further 14 were required for a robust statistical analysis – this gave a total of 24 samples. We have developed a high-content multiplexed approach (a major innovation of this study), where each sample takes a significant amount of time to analyse and produces a large amount of multi-parameter data on a single cell basis (across the 24 specimens, we took 3500 imaging fields containing 2,640,000 cells into our analysis of Vimentin/EpCAM/CD24 co-staining at the tumour-stroma interface). Fields of view were randomly assigned to training and validation sets with a 70:30 split and reported outcomes are based on the validation set. With this high-content multiplexed approach, training and validation of a machine learning approach can be achieved using a relatively small number of specimens.

However, we appreciate the value of this reviewer’s request for a fully independent experimental set, and therefore we have now performed a new experiment staining 60 new specimens (30 metastatic and 30 non-metastatic) as an independent experimental set for the machine learning (Figure 3D). This set was trained and tested on a 70:30 split entirely independently from the first experimental set, and achieved very similar performance. This second set comprised over 9000 imaging fields at the tumour-stroma boundary from 60 evenly stratified tumour specimens, containing over 8.5 million nucleated cells.

The nature of the data in this experiment, with a very large number of datapoints at single cell level, does not lend itself to a multivariate analysis. Machine learning approaches have the major advantage that they can incorporate the complexity of single cell data.

In figure 4, it looks like CD24 is positive in the bulk of tumors (regardless of stage) and in skin. Is this specific? Also, there appear to be VIMENTIN/EPCAM/CD24 positive cells in the bulk of non-metastatic tumours. Can this be seen using sequencing? Overall, the images as presented are not overly convincing.

CD24 on its own is not a specific marker. It is expressed in the epithelial tumour cells, in normal keratinocytes, and in a number of other cell types. Therefore, staining for CD24 would be expected in the tumour bulk and the normal epithelial layer (oral mucosa, in this case). The important advance of our immunofluorescence approach is that we have multiplexed CD24, EpCAM and Vimentin together for identification of co-localisation on a single cell basis. We have added immunofluorescence (Vimentin/EpCAM/CD24) data as a supplementary figure (Figure S1), showing that an EMT sub-population co-staining for Vimentin, CD24 and EpCAM is present in cancer culture but absent in cultured keratinocytes and fibroblasts. However, CD24 and EpCAM staining alone are both abundant in bulk tumour cells and normal keratinocytes and Vimentin is abundant in fibroblasts. It is the multiplexing of the three markers together (combined with image analysis at the single cell level) that is key, and a major innovation of our study.

In addition, we have analysed a scRNAseq dataset for head and neck cancer (Puram et al., 2017), and this showed that co-expression of Vimentin, EpCAM and CD24 in single cells is specific to a subset of tumour cells and not present in stromal cells. This data has been added to the manuscript (Figure S4).

We detected only very few Vimentin/EpCAM/CD24 co-staining cells in the body of non-metastatic tumours (quantification shown in Figure 2F). This was dwarfed by the number seen in the tumour stroma, particularly of metastatic tumours.

The comment that the images presented in figure 4 (now Figure 2) are not overly convincing agrees with more detailed comments from reviewer 2, and we will address this point in detail under reviewer 2’s comments.

EMT stem versus restricted signatures should be validated using additional models.

We agree that this aspect of the manuscript was somewhat preliminary and, on reflection, detracted from the main message of the manuscript – the first identification of individual EMT CSCs disseminating from the tumour in human pathological specimens, which are predictive of metastasis. We have therefore revised our manuscript to focus on these major findings, and have removed this more preliminary work on transcriptional signatures.

Also, greater evidence is required to determine how these cell fractions may differ. Are they sitting in different epigenetic states? Can trajectories be detected in human cancers, using single cell sequencing, for example?

We agree that these are very interesting questions. However, we think they are outside the scope of the current study. We have revised the manuscript to make the scope and major focus of the study clearer.

Finally, do they have different metastatic potentials?

The differing metastatic potentials of EMT-stem and EMT-restricted sub-populations has been previously demonstrated in animal metastasis models. A requirement for sequential EMT and MET in metastasis is well-established (Tsai et al., 2012), and the ability to undergo MET to seed secondary lesions is restricted to a specific stem-like EMT sub-population in metastasis (Ocana et al., 2012; Pastushenko et al., 2018). In addition, we have previously described the differing tumour initiation potential of the EMT-stem and EMT-restricted sub-lines used in our work – EMT-stem has tumour initiation potential when transplanted orthotopically into NOD/SCID mice, and EMT-restricted does not (Biddle et al., 2016). This is on the background of a cell line where we have demonstrated that EMT drives metastasis (Biddle et al., 2011), and relates to metastatic potential as a cell must have tumour initiation potential in order to seed metastasis. These studies provide the underlying rationale for using ability to undergo MET as the criteria for identifying markers of the EMT CSCs for use in the current study.

The aim of the current study is to translate these findings into human specimens, in order that the findings of metastasis driven by EMT CSCs in experimental models can be investigated in human pathological specimens. Again, in revising the manuscript we have been careful to clarify the scope to focus on these major aims.

Reviewer #2 (General assessment and major comments (Required)):The authors tackle the important and intractable question of the mismatch between the primacy of EMT in cell culture studies versus the rarity with which EMT is morphologically apparent in resected tumour tissues.The early part of the study is convincing and well conducted, with identification of subpopulations of EMT cells with the ability to undergo MET, and associated marker profiles in flow cytometry.They then develop an impressive multiplex assay for the identification of cells with the same profile in resected tumour material- a really promising approach bringing molecular findings into the context of primary tumour tissue.The major issue that I have is in the application of this assay to tissues, and the subsequent AI analysis. Only one example of the putative invading population is shown (Figure 4C) and the stromal 'infiltrative' subpopulation is adjacent to a very flat and 'pushing' tumour/stroma boundary, with no apparent budding into the stroma. This would need to be addressed with several more examples and high-magnification H&E images. Furthermore, this is a major claim- namely that occult infiltrating EMT cells are commonly encountered in peritumoural stroma but can only be differentiated from somatic stroma by multiplex immunofluorescence- and it needs major evidence to back it up. What do these cells look like on H&E? Are they mesenchymal in their appearances on H&E? Can they be conclusively differentiated from other stromal constituents (eg myofibroblasts, plasma cells) immunohistochemically and/or morphologically? It could be that the power to predict metastatic status power is related to somatic stromal factors rather than EMT.

We are very pleased that this reviewer has noted both the importance of our research question and the promise of our approach. They also note that our experiments focusing on EMT CSC marker identification and AI prediction of metastasis are compelling. This reviewer notes the single major concern, also noted by reviewer 1, that the images in figure 4 (now figure 2) are not sufficient to support our major finding that occult infiltrating EMT cells are present in human tumours and can be differentiated from somatic stroma using multiplexed EMT CSC markers. We fully appreciate this concern, and agree that a greater range of images in the revised manuscript, along with clarification of some specific points, is required. Specifically:

Only one example of the putative invading population is shown (Figure 4C) and the stromal ‘infiltrative’ subpopulation is adjacent to a very flat and ‘pushing’ tumour/stroma boundary, with no apparent budding into the stroma. This would need to be addressed with several more examples and high-magnification H&E images.

In metastatic tumours, there were a range of tumour/stroma boundary presentations, including both flat and budding boundaries. We chose the image in Figure 4C (now Figure 2C) because the flat boundary is clearly identifiable as a boundary to a broad, non-specialist audience. However, we agree that we need to present more images, and we have now included more examples with different boundary presentations (Figure 2G and Figure S3).

What do these cells look like on H&E? Are they mesenchymal in their appearances on H&E? Can they be conclusively differentiated from other stromal constituents (eg myofibroblasts, plasma cells) immunohistochemically and/or morphologically? It could be that the power to predict metastatic status power is related to somatic stromal factors rather than EMT.

We agree that, in order to support our findings, we need to convincingly demonstrate that we are staining infiltrating tumour cells and not stromal constituents. In support of this conclusion, we can make a couple of observations based on the data already included. (1) There is no co-staining in the stroma of normal mucosal regions that are distant from the tumour. (2) Our use of EpCAM as a tumour lineage marker is specifically intended to exclude staining for stromal constituents. EpCAM is a specific epithelial marker, that is not expressed in stromal or immune cells – it is expressed exclusively in epithelia and epithelial-derived tumours (Keller et al., 2019), and we have made this point more clearly in the revised manuscript. Contrastingly, CD24 is quite widely expressed, and Vimentin is expressed by stromal fibroblasts. They are only specific to the EMT CSCs when multiplexed together with EpCAM.

Analysis of H&Es proved challenging, as the immunostained sections are often several sections away from the H&Es. This can mean that there is up to 100µm between them and, whilst the broad structures are preserved, the individual cells are different. Due to this, we were unable to identify the triple-stained cells in the H&Es. Instead, to further test the conclusion that we are staining infiltrating tumour cells and not stromal constituents, we analysed a published scRNAseq dataset for head and neck cancer (Puram et al., 2017). This showed that co-expression of Vimentin, EpCAM and CD24 in single cells is specific a subset of tumour cells and not present in stromal cells. This data has been added to the manuscript (Figure S4). This strengthens our conclusion that we are staining infiltrating tumour cells and not stromal constituents.

The AI prediction of metastatic status is compelling, but this fundamental point would need to be persuasively addressed in order to support the author's major claims.I do not feel qualified to comment upon the AI strategies used later in the study.Reviewer #3 (General assessment and major comments (Required)):The article by Youssef G et al., focused on developing a Machine Learning system to use immunofluorescence data to detect metastatic cells in tumor stroma, which might be responsible for metastasis in case of OSCC. To detect single cell in the transition of EMT to MET they focused on EMT-Stem cells rather than only EMT phenotypes. They have shown that retention of epithelial marker EpCAM and stem cell marker CD24 and upregulation mesenchymal marker Vimentin can identify disseminating EMT stem cells in the tumor stroma. It is very well presented, well written and has high implication.

We are grateful for this reviewer’s very positive comments on both the importance and presentation of our study, as well as their useful suggestions for improving the manuscript.

Comments to improve:1. Strongly recommended to add the distribution of tumor status vs. proposed marker expression pattern. That is to show the distribution of EpCAM, CD24, Vimentin +/- in metastatic vs. other tumor status as mentioned in Supplementary figure 2. This might help you to establish these markers combination to follow a pattern in disease progression.

The nature of the data in this experiment, with a very large number of datapoints at single cell level, does not lend itself to a multivariate analysis. Machine learning approaches have the major advantage that they can incorporate the complexity of single cell data.

2. In all cell and tissue images add the scale.

This has now been added.

3. For figure 3f, show enlarged picture of the single cell staining on the inset or add a separate panel to show only single cell staining.

This has now been added (now Figure 1F).

4. Figure 4, the panel name or the font is too small to read, enlarge the font size (a, b, c, d, f).

All font sizes have now been enlarged and checked for readability.

5. Same problem with figure 6a, font size too small. In addition, in the heat maps, is it possible to add cluster names horizontally? Also for figure 6c, the cluster names are too small.

Figure 6 has now been removed, in order to ensure a coherent focus on the main message of the manuscript – the first identification of individual EMT CSCs disseminating from the tumour in human pathological specimens, which are predictive of metastasis.

6. The EMT sub-populations are not associated with a spectrum of epithelial/mesenchymal genes expression (supplementary figure 5). The explanation is not very clear.

This has also been removed, for the same reason.

[Editors’ note: what follows is the authors’ response to the second round of review.]

The reviewers have discussed their reviews with one another, and the Reviewing Editor has drafted this to help you prepare a revised submission.Essential revisions:1) The authors should test the staining-AI protocol use another independent blinded cohort.

We do not have the capacity to add a third cohort to this study. However, we have performed a blinded analysis of the second cohort. This blinded analysis is now described in the paper; it involved using the ANN trained using the first cohort to predict the metastatic status of the second cohort with the researchers blinded, prior to provision of clinical data by the pathologist. In the original revised manuscript, we omitted this analysis as we thought it better to treat the two cohorts separately. However, having reflected on the reviewers’ comments we see that this was a mistake. This blinded prediction is important evidence supporting predictive accuracy of the immunofluorescent signature, so is now included in the manuscript.

We have also added the following statement: ‘To support the conclusions of this study, a further blinded cohort and a greater number of tumour specimens would be valuable.’

2) Please add clinical data for all patient samples.

Clinical data for cohort 1 is in supplementary file 1. Clinical data for cohort 2 is now added as supplementary file 2.

3) The authors should edit manuscript to ensure clarity.

We have edited the manuscript to improve clarity, ensure a focussed narrative, and explain unfamiliar terms.

4) The need of the AI approach should be better articulated.

To answer this question, we compared our image-based AI approach with a simpler approach where an ANN is trained using summary data of positive cell counts (Figure 3E). This latter approach is similar to a logistic regression, and had poor predictivity. This supports the importance of an image-based AI approach for prognostic prediction.

Reviewer #2 (Recommendations for the authors):Here author used cell lines to standardize their protocol and used a test patient set to validate these markers' potential. Here they used a set of specimen slides with known histopathology and checked the marker's potential to predict the tumor status and how efficient the technology was.– The major criticism is that the authors did not test the AI to determine the tissue status from a blind sample set. They need to show the efficiency of the AI to determine tumor, non-tumor, or metastasis status from a pool of specimens and match the AI predictions with the standardized histopathology data and their four markers IF protocols. What is the level of detection efficiency and what is the sensitivity and specificity of the technology?– While validating a new marker for OSCC, even with a small sample set, it is strongly recommended to show metastasis potential in vivo.

The method for applying tumour mask is displayed in Figure 2E for cohort 1 and Figure 2—figure supplement 3 for cohort 2. Briefly, in the image analysis pipeline, dense areas of EpCAM+ (cohort 1) or Vimentin+ (cohort 2) cells are merged to specify tumour/stroma regions. Thus, CSCs inside tumours (in the EpCAM-dense tumour region) can be discriminated from CSCs invading the surroundings (in the Vimentin-dense stromal region).

Reviewer #3 (Recommendations for the authors):The paper has numerous technical issues which will have to be addressed before accepting it. First and foremost, I am baffled why the fibroblasts in Figure S1 are EpCAM positive. If this is not an error, I can't understand the rest of the manuscript.Beyond that, I find many of the figures challenging to the eye and in the case of Figure 2F, making key evidence hard to understand. Perhaps because of that, I couldn't see anywhere explaining properly how tumour "mask" was applied (Figure 2E?!).Figure 3 also needs revision: none of the terms is described anywhere. For non-expert biologists reading eLife terms such as Epochs are entirely incomprehensible.

The fibroblasts in Figure 1—figure supplement 1 are in fact negative for EpCAM (in red), whereas the normal keratinocytes are negative for Vimentin (yellow). Only the tumour cells possess all three markers, and contain some cells that are positive for all three.